# Scaling Speech-Text Pre-training with Synthetic Interleaved Data

**Aohan Zeng[‡§∗], Zhengxiao Du[‡§∗], Mingdao Liu[‡∗], Lei Zhang[§], Shengmin Jiang[§]**
**Yuxiao Dong[‡], Jie Tang[‡]**
[‡]Tsinghua University    [§]Zhipu.AI
Code & Models: `https://github.com/THUDM/GLM-4-Voice`

## Abstract

Speech language models (SpeechLMs) accept speech input and produce speech output, allowing for more natural human-computer interaction compared to text-based large language models (LLMs). Traditional approaches for developing SpeechLMs are constrained by the limited availability of unsupervised speech data and parallel speech-text data, which are significantly less abundant than text pre-training data, thereby limiting their scalability as LLMs. We propose a novel approach to scaling speech-text pre-training by leveraging large-scale synthetic interleaved data derived from text corpora, eliminating the need for parallel speech-text datasets. Our method efficiently constructs speech-text interleaved data by sampling text spans from existing text corpora and synthesizing corresponding speech spans using a text-to-token model, bypassing the need to generate actual speech. We also employ a supervised speech tokenizer derived from an automatic speech recognition (ASR) model by incorporating a vector-quantized bottleneck into the encoder. This supervised training approach results in discrete speech tokens with strong semantic preservation even at lower frame rates (e.g. 12.5Hz), while still maintaining speech reconstruction quality. Starting from a pre-trained language model and scaling our pre-training to 1 trillion tokens (with 600B synthetic interleaved speech-text data), we achieve state-of-the-art performance in speech language modeling and spoken question answering, improving performance on spoken questions tasks from the previous SOTA of 13% (Moshi) to 31%. We further demonstrate that by fine-tuning the pre-trained model with speech dialogue data, we can develop an end-to-end spoken chatbot that achieves competitive performance comparable to existing baselines in both conversational abilities and speech quality, even operating exclusively in the speech domain.

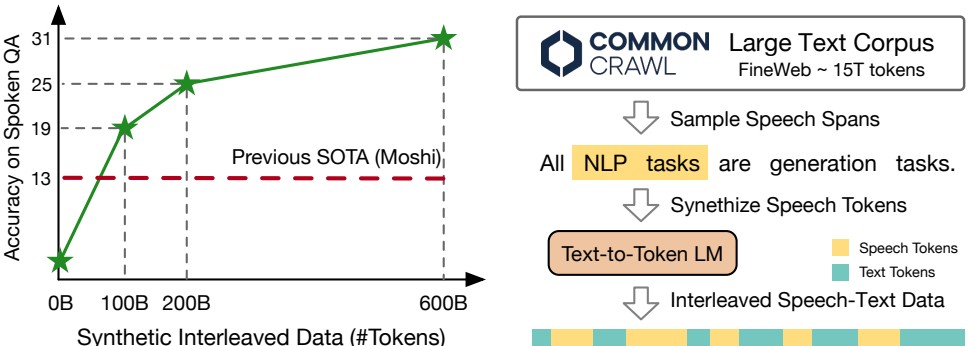

Figure 1: (Left) The performance on Spoken QA continuously improves as the amount of synthetic interleaved data increases, significantly surpassing the previous SOTA (Moshi). (Right) The pipeline for synthesizing interleaved speech-text data.

---

∗Equal contribution. Email: {`zah22,zx-du20,liumd24`}`@mails.tsinghua.edu.cn`
§Work was done when ML, LZ interned at Zhipu.AI. Corresponding authors: YD and JT.

# 1 INTRODUCTION

Large language models (LLMs) have significantly advanced natural language processing, demonstrating capabilities beyond traditional language tasks. Trained on vast internet corpora, they exhibit emergent abilities such as instruction following (Ouyang et al., 2022), logical reasoning (Wei et al., 2022), and tool utilization (Schick et al., 2023). These advancements have enabled applications like interactive chatbots and personalized digital assistants. However, an ideal AI assistant should not rely solely on text. Voice-based interaction offers a more natural and intuitive interface for human-AI interaction. Traditional voice-based systems combine Automatic Speech Recognition (ASR), LLMs, and Text-to-Speech (TTS) models in a cascading manner. This approach, however, suffers from information loss during ASR and TTS processes, limiting the ability to capture and express the rich nuances of speech.

Speech language models (SpeechLMs) have emerged as a promising approach for building general-purpose voice assistants capable of processing speech input and output end-to-end. Several methods have been explored to construct SpeechLMs. Lakhotia et al. (2021) proposed unsupervised learning on speech corpora using discrete semantic tokens. Hassid et al. (2023) improved performance by initializing from pre-trained language models, while Moshi (Défossez et al., 2024) utilized large-scale training on private speech data. However, a key challenge remains: the scarcity of speech data compared to text data. While text corpora like FineWeb (Penedo et al., 2024) offer 15 trillion high-quality tokens, large unsupervised speech datasets like VoxPopuli (Wang et al., 2021) provide only 400K hours of speech, equivalent to 36 billion tokens at 25Hz. This disparity limits the scalability and performance of SpeechLMs relative to LLMs.

A straightforward idea to address this limitation is to synthesize speech from text pre-training corpora using TTS models. However, this approach faces three major challenges. First, the lower information density of speech tokens leads to significant token expansion, drastically reducing training efficiency. Second, the process of synthesizing speech for large-scale text corpora is computationally expensive. Third, training on pure speech data fails to align with the text modality, preventing the model from leveraging the capabilities of existing LLMs. Recently, Nguyen et al. (2024) has explored the use of *interleaved speech-text data* for training. This approach improves alignment between speech and text modalities, leading to better speech language modeling performance. However, their method requires parallel speech-text datasets to construct the interleaved data, which significantly limits its scalability for large-scale pre-training.

In this paper, we propose a novel approach to scaling speech-text pre-training by synthesizing interleaved speech-text data from text corpora. The interleaved data is generated by sampling text spans and converting them into speech tokens using a text-to-token model. This efficient process bypasses the need to generate actual speech, enabling large-scale pre-training without relying on extensive speech datasets. Inspired by Du et al. (2024), we train the tokenizer in a supervised manner using ASR models and datasets. Experiments with sampling rates from 6.25Hz to 50Hz revealed trade-offs between semantic retention, model efficiency, speech reconstruction quality, and pre-training performance. We selected 12.5Hz as the optimal rate for balancing these factors. To synthesize large-scale interleaved data, we used existing TTS datasets to train a text-to-token model, generating 600B tokens of interleaved speech-text data and expanding the pre-training to 1 trillion tokens. Finally, through fine-tuning on speech dialogue data, we developed an end-to-end spoken chatbot operating entirely in the speech domain. The main contributions of this paper are as follows:

- We propose a novel method to effectively synthesize high-quality interleaved speech-text data from text corpora, addressing data limitation challenges in speech-text pre-training.
- We design a SpeechLLM architecture featuring a 12.5Hz single-codebook speech tokenizer trained in a supervised manner, along with a flow-matching based decoder for speech reconstruction, achieving both robust semantic preservation and high-quality speech synthesis.
- We scale our pre-training to 1 trillion tokens using synthesized interleaved speech-text data, significantly advancing capabilities in speech language modeling and spoken question answering.
- We develop an end-to-end spoken chatbot by fine-tuning pre-trained models with speech dialogue data, achieving competitive performance in conversational abilities and speech quality while operating exclusively in the speech domain.

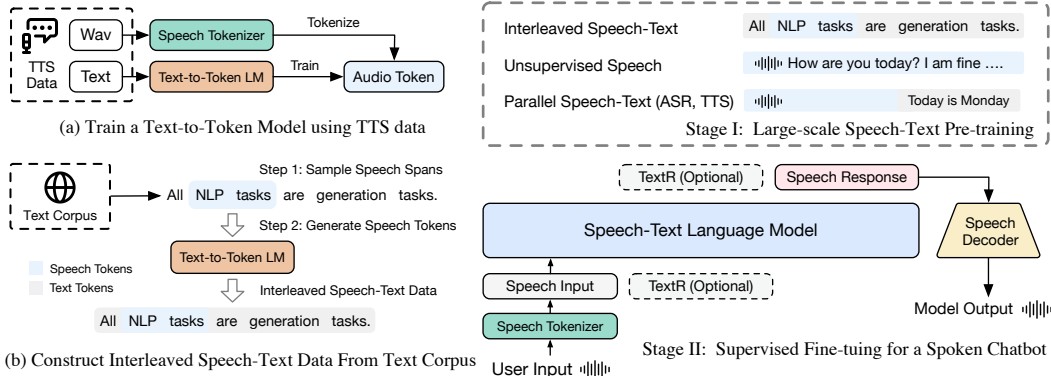

(a) Train a Text-to-Token Model using TTS data

(b) Construct Interleaved Speech-Text Data From Text Corpus

Figure 2: **Overview of our method.** First we train a text-to-token model to construct interleaved speech-text data. The speech language model's training contains two stages. In the stage 1 the model is pre-trained with synthetic speech-text interleaved data. In the stage 2 the the model is fine-tuned with a speech dialogue dataset.

## 2 OUR APPROACH

Current approaches for build SpeechLMs typically fall into two categories. One method (Fang et al., 2024; Défossez et al., 2024) involves the language model for speech input but outputs embeddings for an additional non-autoregressive (NAR) model to generate speech tokens, which limits the modeling capacity and potentially reduces the upper bound of performance. The other method (Xie & Wu, 2024) uses inconsistent audio representations for input and output, leading to misalignment between input and output modalitiy.

In this section, we present our approach for developing an end-to-end spoken chatbot using a unified speech-text modeling framework. Our method integrates a supervised speech tokenizer, a technique for synthesizing interleaved speech-text data, and a two-stage training process to extend pre-trained language models to the speech domain. This comprehensive approach enables us to leverage large-scale text data for speech modeling, effectively aligning speech and text modalities within a single model.

### 2.1 SPEECH TOKENIZATION

**Supervised Speech Tokenizer** Previous methods of discrete speech tokenizers are either trained with reconstruction/adversarial objectives of speech waveform (Wang et al., 2023; Chen et al., 2024) or self-supervised learning on automatically discovered acoustic units(Hsu et al., 2021). Following recent advance in text-to-speech synthesis (Du et al., 2024), we train the discrete speech tokenizer by fine-tuning a pretrained automatic speech recognition (ASR) model with an additional pooling layer and a vector quantization layer in the middle of the encoder.

The pooling layer is a 1D average pooling operator of window size $k$, which reduces the sampling rate to a fraction of $1/k$. The vector quantization layer approximates the continuous intermediate representations in the encoder with the closest vectors in the codebook. The selected indices in the codebook are used as the speech token indices. The codebook vectors are learned with exponential moving average (EMA) and we add a commitment loss to restrict the volume of continuous representations before quantization. To overcome codebook collapse, we apply the random restart trick (Dhariwal et al., 2020) to reset vectors whose mean usage falls below a certain threshold.

We also adapt the Whisper architecture to support streaming inference, which is important to reduce latency for online speech interaction. We replace the convolution layer before the encoder Transformer with the causal convolution layer (van den Oord et al., 2016). We also replace the bidirectional attention in the encoder with block causal attention: the input audios are divided into segments of equal intervals and positions in a segment and attend to all the positions in the current segment and previous segments, but not positions in the following segments. Empirically we set the segment interval to 2 seconds (100 tokens before the average pooling). We find this can match

Table 1: **Speech Reconstruction Results.** We evaluate semantic retention with Word Error Rate (WER) and reconstruction quality with VisQOL (Hines et al., 2015) and MOSNet (Lo et al., 2019) for different speech tokenizers across various frame rates. The baseline results are independently evaluated by us.

| Model | Frame Rate | Bitrate | Causal | LibriSpeech | | |
|---|---|---|---|---|---|---|
| | | | | WER↓ | VisQOL↑ | MOSNet↑ |
| Ground Truth | - | - | - | 4.62 | - | 3.27 |
| RVQGAN | 75Hz | 1.50K | ✗ | - | 1.74 | 2.74 |
| SemantiCodec | 50Hz | 1.30K | ✗ | - | 2.43 | 3.12 |
| SpeechTokenizer | 50Hz | 1.50K | ✗ | - | 1.53 | 2.67 |
| SpeechTokenizer | 50Hz | 4.00K | ✗ | - | 3.07 | 3.10 |
| Spirit-Base | 25Hz | 225.0 | ✗ | 11.66 | - | - |
| Spirit-Expressive | 38.5Hz | 307.0 | ✗ | 10.60 | - | - |
| Moshi (Mimi) | 12.5Hz | 1.10K | ✓ | 8.36 | 2.82 | 2.89 |
| Ours | 50Hz | 600 | ✓ | 6.24 | 2.67 | 3.38 |
| | 25Hz | 300 | ✓ | 6.80 | 2.60 | 3.33 |
| | 12.5Hz | 175 | ✓ | 8.43 | 2.52 | 3.39 |
| | 6.25Hz | 100 | ✓ | 14.41 | 2.34 | 3.24 |

the ASR performance of bidirectional attention. For more details about speech tokenizer training, please refer to Appendix B.1.

**Speech Decoder**  Given discrete speech tokens, we synthesize speech through the speech decoder. We follow the decoder architecture of CosyVoice (Du et al., 2024), which consists of a speech token encoder, a conditional flow matching model (Mehta et al., 2024), and a HiFi-GAN vocoder (Kong et al., 2020). The speech token encoder converts a sequence of discrete tokens into a sequence of contextual vectors with a Transformer encoder. To facilitate the streaming synthesis of speech, we adapt the speech token encoder to use the same block causal attention as the speech tokenizer. The flow matching model generates Mel spectrograms conditioned on the speech token representations. Finally, the generated Mel spectrograms converted into the speech waveforms through the HiFi-GAN vocoder (Kong et al., 2020). To train the speech decoder, we use the unsupervised speech data described in Section 2.3.1, which consists of various speakers. For more details about speech decoder training, please refer to Appendix B.2.

We evaluate the content preservation and quality of generated speech by our speech decoder on LibriSpeech (Panayotov et al., 2015). The results are shown in Table 1. We measure the content preservation by the Word Error Rate (WER) between the transcription with an ASR model provided in Nguyen et al. (2023) and the true transcription. For speech quality, following Défossez et al. (2024), we compute VisQOL (Hines et al., 2015) and MOSNet (Lo et al., 2019) of the reconstructed speech. Our tokenizer performs well across various sampling rates, with the 12.5Hz variant offering an optimal balance between efficiency and quality. It maintains high quality scores (MOSNet 3.39) and content preservation (WER 8.43) while significantly reducing bitrate (52.7). Our ablation study on sampling rates during pre-training (Cf. Section 3.3.2) shows that lower rates improve performance, but gains plateau at 12.5Hz. Based on these results, we select the 12.5Hz variant for our subsequent experiments.

## 2.2 SYNTHESIZE INTERLEAVED SPEECH-TEXT DATA

Interleaved speech-text data consists of tokens where speech and text sequences are interleaved at the word level. For example, a sequence might look like: "Today is `<Speech_24>` `<Speech_5>` ... `<Speech_128>` day". We hypothesize that training on interleaved speech-text data encourages the model to learn an alignment between speech and text, facilitating the transfer of text-based knowledge to speech representations. Previous methods for creating interleaved speech-text data rely on aligned speech-text parallel datasets (Nguyen et al., 2024), which are challenging to obtain. We propose a novel and efficient approach for constructing interleaved speech-text data using existing text

datasets. The process consists of two main steps. First, we train a text-to-token model that directly converts text into corresponding speech tokens, eliminating the need to synthesize actual speech. This approach avoids the error accumulation associated with text-to-speech-to-token pipelines and significantly improves synthesis efficiency, making it practical and scalable for large-scale data generation. Next, we sample text spans from existing text datasets and transform them into speech spans using the trained text-to-token model. This enables the efficient and scalable creation of interleaved speech-text data without requiring aligned speech-text parallel datasets.

**Text-to-Token Model**    We train a 1.5B text-to-token model based on standard transformer architecture to convert text into corresponding speech tokens. While these tokens can be further synthesized into actual speech using our speech decoder, this step is unnecessary for constructing interleaved speech-text data. To prepare the training data, we first tokenize speech from text-to-speech datasets into discrete speech tokens. The text-to-token model is then trained to predict these speech token sequences based on the input text. The training objective is to minimize the negative log-likelihood of the predicted speech tokens conditioned on the corresponding text input:

$$\mathcal{L} = -\sum_{i=1}^{N} \sum_{j=1}^{M_i} \log P(a_{i,j} | T_i, a_{i,<j}; \theta) \tag{1}$$

where $T_i$ is the i-th input text, $a_{i,j}$ is the j-th audio token in the i-th sample, $M_i$ is the length of the i-th speech token sequence, $\theta$ represents the model parameters, and $N$ is the number of training samples.

We use a multi-speaker text-to-speech datasets to train this model (see Appendix A.3 for detailed data distribution). We also include additional high-quality text-speech pairs generated using the CosyVoice (Du et al., 2024) model to improve accuracy for short or incomplete text spans. The architecture and the training details

Table 2: **Word Error Rate (WER) of the text-to-token model**. The dataset labeled "Interleaved Data" refers to text and speech pairs generated during the construction of interleaved speech-text data.

| Dataset | Language | WER (%) |
|---|---|---|
| VCTK | English | 3.20 |
| Interleaved Data | English | 11.62 |
|  | Chinese | 9.34 |

about the text-to-token model training can be found in Appendix B.3. To speedup the speech token generation process, we deployed the model using the SGLang framework (Zheng et al., 2024), achieving a generation speed of 25k tokens per second on a single H800 instance.

**Interleaved Data Construction**    To construct interleaved data from a text document, we apply a span corruption technique that randomly selects spans from the text sequence. Span lengths are drawn continuously from a Poisson distribution ($\lambda = 10$) until the total length of selected spans reaches the predefined ratio $\eta$ of the original text length. Next, text spans corresponding to the drawn lengths are randomly selected from the document. These spans are converted into speech tokens using the text-to-token model, producing an interleaved speech-text sequence. The span corruption ratio $\eta$ plays a crucial role in enabling effective knowledge transfer between the speech and text modalities, as demonstrated in our ablation study (Cf. Section 3.3.3). Based on the findings of this study, we set $\eta$ to 0.3 for optimal performance. We selected high quality text datasets (FineWeb-Edu (Penedo et al., 2024) for English and Chinese-Fineweb-Edu (OpenCSG Community) for chinese) to apply the previously mentioned synthesis process, generating a total of 600B tokens, with a 2:1 ratio of English to Chinese. Appendix D.3 provides samples of interleaved data constructed.

We evaluated the performance of the text-to-token model on the VCTK (Yamagishi et al., 2019) dataset and the interleaved data using word error rate (WER) as the evaluation metric. To compute WER, we used our speech decoder to synthesize real speech from the speech tokens generated by the text-to-token model, and then transcribed using `whisper-large-v3` (Radford et al., 2023). The results are summarized in Table 2. For the standard VCTK dataset, we observed a lower WER of 3.20. However, the WER for speech spans generated from the text pre-training data was higher. We attribute this discrepancy primarily to some spans in the text pre-training data being difficult to pronounce accurately.

## 2.3 TRAINING

We initialize the speech language model with a pre-trained large language models' parameters to leverage its existing knowledge. In order to support speech processing, we extend the model's vocabulary and embedding space. Specifically, we augment the original language model vocabulary, $V_{\text{lang}}$, with a discrete speech vocabulary, $V_{\text{speech}}$, resulting in a combined vocabulary, $V = V_{\text{lang}} \cup V_{\text{speech}}$. This expansion allows the model to accept both text and speech tokens as input and output. However, the ability to effectively understand and generate these tokens relies on subsequent training to align the text and speech modalities.

The training process consists of two stages. In the first stage, the model is pre-trained on synthetic interleaved data to learn the alignment between text and speech. In the second stage, fine-tuning is performed using speech dialogue data to enable the model to handle speech interactions.

### 2.3.1 SPEECH-TEXT PRE-TRAINING

To extend LLMs' capabilities in speech-text tasks, we introduce a speech-text pre-training stage. This stage enables the model to process and represent discrete speech tokens. We utilize four data types, each serving a specific purpose:

- **Interleaved speech-text data:** As described in Section 2.2, these datasets facilitate cross-modal knowledge transfer between text and speech.
- **Unsupervised text data:** We use a diverse corpus, similar to GLM et al. (2024), containing 10T tokens from webpages, Wikipedia, books, code, and research papers to maintain the model's language understanding.
- **Unsupervised speech data:** Using the Emilia pipeline (He et al., 2024), we collected 700k hours of high-quality English and Chinese speech data, filtered by DNSMOS P.835 scores above 2.75, ensuring diverse and clean speech inputs.
- **Supervised speech-text data:** This includes ASR and TTS data, teaching the model to learn bidirectional relationships between speech and text.

Both text and speech are represented as discrete tokens. The model is trained the next-token prediction objective with cross-entropy loss function. For text, speech, and interleaved data, the model is trained to predict all the tokens. For supervised speech-text data, the model is only trained to predict tokens in the target parts (text in ASR data and speech in TTS data). We set text data at 30% of each batch to preserve language ability. Unsupervised speech and supervised speech-text data were trained for one epoch each, while interleaved data filled the remaining capacity, balancing language comprehension and speech processing. The detailed training data distribution can be found in Appendix A.

### 2.3.2 SUPERVISED FINE-TUNING

Following speech-text pre-training, we fine-tune our model for speech dialogue tasks using a dataset derived from Magpie (Xu et al., 2024). We use GPT-4 to adapt the original text-based dialogues for speech scenarios by filtering examples, shortening responses, and avoiding outputting text that cannot be read aloud. The detailed prompt for this adaptation process can be found in the Appendix E. This curation process results in our SpeechDialog-90K dataset, which contains 90K triplets (SpeechI, TextR, SpeechR), where SpeechI is the speech instruction, TextR is the text response, and SpeechR is the corresponding speech response synthesized from TextR using MeloTTS (Zhao et al., 2023). For train hyper-paramaters, we use a batch size of 64, a sequence length of 4096 tokens, and train for 10 epochs with a learning rate decaying from 5e-5 to 5e-6. We use the AdamW optimizer.

## 2.4 INFERENCE

During inference, our framework supports two modes: speech-to-speech and text-guided speech generation. In speech-to-speech mode, the model directly processes speech input to generate speech output. Our streaming tokenizer converts the user's speech into discrete tokens, which the model then processes to generate output speech tokens. These output tokens are synthesized into continuous speech by our block-wise decoder, operating on 2-second blocks (25 tokens at 12.5Hz). The

Table 3: **Pre-training Results.** 'S': speech input and output. 'S→T': speech input and text output. 'T→S': text input and speech output. Results for Spirit-LM are taken from Nguyen et al. (2024) and other results are from Défossez et al. (2024). We use ∅ to indicate tasks and modalities not supported by the model, and - to indicate scores that are not publicly available.

| Model | Speech Language Modeling | | | | | | Spoken Question Anwsering | | | | | |
| | sTopic-StoryCloze | | | sStoryCloze | | | Web Questions | | Llama Questions | | TriviaQA | |
| | S | T→S | S→T | S | T→S | S→T | S | S→T | S | S→T | S | S→T |
|---|---|---|---|---|---|---|---|---|---|---|---|---|
| GSLM | 66.6 | ∅ | ∅ | 53.3 | ∅ | ∅ | 1.5 | ∅ | 4.0 | ∅ | - | - |
| AudioLM | - | ∅ | ∅ | - | ∅ | ∅ | 2.3 | ∅ | 7.0 | ∅ | - | - |
| TWIST | 76.4 | ∅ | ∅ | 55.4 | ∅ | ∅ | 1.1 | ∅ | 0.5 | ∅ | - | - |
| Spirit-LM | 82.9 | 72.7 | 88.6 | 61.0 | 59.6 | 64.6 | - | - | - | - | - | - |
| SpeechGPT | ∅ | ∅ | ∅ | ∅ | ∅ | ∅ | ∅ | 6.5 | ∅ | 21.6 | ∅ | 14.8 |
| Spectron | - | - | - | - | - | - | ∅ | 6.1 | ∅ | 21.9 | ∅ | - |
| Moshi | **83.0** | ∅ | ∅ | 60.8 | ∅ | ∅ | 9.2 | 26.6 | 21.0 | 62.3 | 7.3 | 22.8 |
| Ours (9B) | 82.9 | **85.0** | **93.6** | **62.4** | **63.2** | **76.3** | **15.9** | **32.2** | **50.7** | **64.7** | **26.5** | **39.1** |

block-wise encoder computes representations which are then used by the conditional flow matching model to generate Mel spectrograms. Finally, these spectrograms are converted into continuous speech using the HiFi-GAN vocoder. For text-guided speech generation, given the speech input (SpeechI), the model generates both a text response (TextR) and the corresponding speech response (SpeechR) in a single forward pass. The text response is generated as an intermediate step, which then guides the production of the final speech output. The corresponding template for two modes can be found in appendix F.

## 3 EXPERIMENTS

### 3.1 EXPERIMENTAL SETUP

**Configuration** We employ `GLM-4-9B-Base` (GLM et al., 2024) as our base LLM for experiments. For ablation, we also use a smaller LLM with 1.5 billion parameters detailed in Table 13. Our speech-text pre-training stage processes a total of 1T tokens, with a fixed sampling of 30% text data, one epoch each of unsupervised speech and supervised speech-text data, and the remainder consisting of interleaved data. Throughout the pre-training stage, we maintain a sequence length of 8192 tokens and use a learning rate that linearly decays from 6e-5 to 6e-6. For the fine-tuning phase, we use a batch size of 64, a sequence length of 4096 tokens, and train for 10 epochs on the fine-tuning dataset with a learning rate decaying from 5e-5 to 5e-6. We use the AdamW optimizer for both pre-training and fine-tuning stages.

**Baselines** For pre-trained models, We compare our method with GSLM (Lakhotia et al., 2021), AudioLM (Borsos et al., 2023), TWIST (Hassid et al., 2023), Spirit-LM (Nguyen et al., 2024), SpeechGPT (Zhang et al., 2023), Spectron (Nachmani et al., 2024), and Moshi (Défossez et al., 2024). Except GSLM and AudioLM, other baselines are based on a text pretrained language model. Note that Moshi is pretrained on a speech collection of 7 million hours, an order of magnitude larger than our unsupervised speech data. For chat, we only compare end-to-end spoken chatbots supporting speech as both input and output, we choose SpeechGPT (Zhang et al., 2023), Llama-Omni (Fang et al., 2024), Mini-Omni (Xie & Wu, 2024) and Moshi (Défossez et al., 2024). Moshi is fine-tuned for full duplex conversations and each conversation must begin with a greeting from the model. Therefore, we wait 3 seconds for the greeting to end before asking the speech query. For Mini-Omni, we use the default AT mode for evaluation.

**Speech Language Modeling** We first evaluate the pretrained model's ability to model speech by the accuracy of selecting the correct continuation of a given context according to the predicted likelihood. We consider three different settings: from speech context to speech continuation (denoted as 'S'), from text context to speech continuation (denoted as 'T→S'), and from speech context to text continuation (denoted as 'S→T'). We use two datasets proposed by Hassid et al. (2023), Spoken

Table 4: **Evaluation results for end-to-end spoken chatbots.** All baseline results in this table were obtained through our own evaluation.

| | w/o Text | Content Quality | | Speech Quality | |
|---|---|---|---|---|---|
| | | General QA↑ | Knowledge↑ | UTMOS↑ | ASR-WER↓ |
| SpeechGPT | ✗ | 1.40 | 2.20 | 3.86 | 66.57 |
| Mini-Omni | ✗ | 2.44 | 1.10 | 3.17 | 25.28 |
| Llama-Omni | ✗ | 3.50 | 3.90 | 3.92 | 9.18 |
| Moshi | ✗ | 2.42 | 3.60 | 3.90 | 7.95 |
| 9B + Text-guided | ✗ | **3.69** | **4.70** | **4.33** | 7.83 |
| - No Interleaving | ✗ | 2.48 | 1.00 | 4.31 | 10.34 |
| - No Pre-training | ✗ | 2.20 | 0.90 | 4.32 | **6.11** |
| 9B | ✓ | 3.18 | 3.20 | 4.33 | ∅ |
| - No Interleaving | ✓ | 1.21 | 0.01 | 4.33 | ∅ |
| - No Pre-training | ✓ | 1.10 | 0.00 | 4.32 | ∅ |

StoryCloze and Spoken TopicStoryCloze. The baseline results are taken from Nguyen et al. (2024); Défossez et al. (2024).

**Spoken Question Answering**   Similar to closed-book question answering in NLP, spoken question answering requires the speech-language model to answer spoken questions about broad factual knowledge without access to external knowledge. We consider two settings for the model: from spoken questions to spoken answers (denoted as 'S'), and from spoken questions to textual answers (denoted as 'S→T'). We evaluate our model on 3 datasets in Défossez et al. (2024): Web Questions (Berant et al., 2013), Llama Questions (Nachmani et al., 2024), and TriviaQA (Joshi et al., 2017). The baseline results are taken from Défossez et al. (2024).

**Evaluating Spoken Chatbots**   To evaluate the spoken chatbot's capabilities we select two aspects: general question answering and knowledge. For general question answering, we utilized prompts from AlpacaEval's (Li et al., 2023b) `helpful_base` and `vicuna` categories, which are more suitable for voice interactions. The knowledge assessment drew 100 questions from Web Questions, Llama Questions, and TriviaQA datasets. The generated speech was transcribed into text using `whisper-large-v3`, and GPT-4 was used to score the responses on a scale of 1 to 10, with the detailed prompt provided in Appendix G. Additionally, we measured ASR-WER to assess the alignment between generated speech and text, as well as UTMOS (Saeki et al., 2022) to evaluate overall speech quality following Fang et al. (2024).

## 3.2   MAIN RESULTS

The evaluation results for the pretrained model are shown in Table 3. On speech language modeling, our method outperforms baselines on all the tasks except the 'S' setting of spoken Topic-StoryCloze, on which our model achieves comparable accuracy to SpiRit-LM and Moshi. Compared with SpiRit-LM, our method achieves significant improvements on the 'T→S' and 'S→T' setting, indicating that our synthetic interleaved data effectively aligns text and speech modalities. On spoken question answering, our method significant outperforms all the baselines on both the 'S' and 'S→T' setting of three datasets. The improvements are especially substantial on the speech-to-speech setting. On Llama Questions, our method considerably reduces the previous gap between the speech-to-speech and speech-to-text settings, indicating that it effectively transfers the knowledge in the text modality to the speech modality. Overall, our method achieves better performance than the best baseline Moshi, with only a tenth of Moshi's natural speech data.

Table 4 shows the evaluation results for spoken chatbots. Our 9B text-guided model outperforms all baseline models in general question-answering and knowledge-based tasks. It also achieves better results in speech quality evaluation compared to others. Notably, even without text guidance, the 9B model still performs comparably with text-guided baselines, highlighting our method's effectiveness in aligning text and speech modalities.

Table 5: **Ablation study on interleaved data scaling and pre-training data composition.** 'S': speech input and output. 'S→T': speech input and text output. 'T→S': text input and speech output. "Web Q." stands for Web Questions. "Llama Q." stands for Llama Questions.

| Model | Speech Language Modeling | | | | | | Spoken Question Anwsering | | | | | |
| | sTopic-StoryCloze | | | sStoryCloze | | | Web Q. | | Llama Q. | | Trivia QA | |
| | S | T→S | S→T | S | T→S | S→T | S | S→T | S | S→T | S | S→T |
|---|---|---|---|---|---|---|---|---|---|---|---|---|
| 9B (600B Interleave) | **82.9** | **85.0** | **93.6** | **62.4** | **63.2** | **76.3** | **15.9** | **32.2** | **50.7** | **64.7** | **26.5** | **39.1** |
| - No Interleaving | 72.8 | 53.3 | 53.3 | 51.7 | 51.4 | 53.7 | 0.1 | 0.3 | 2.3 | 2.3 | 0.2 | 0.5 |
| - 100B Interleave | 80.9 | 83.6 | 93.3 | 59.4 | 61.3 | 73.4 | 9.3 | 25.4 | 37.0 | 60.0 | 11.7 | 26.9 |
| - 200B Interleave | 82.1 | 84.7 | 93.2 | 61.5 | 62.6 | 76.0 | 13.3 | 29.7 | 44.0 | 63.0 | 18.7 | 31.3 |
| 1.5B | 77.5 | 81.4 | **90.1** | 55.4 | 58.6 | 64.0 | 5.4 | **17.6** | 18.3 | **42.7** | 4.6 | 15.6 |
| - No Interleaving | 74.6 | 55.3 | 51.3 | 51.7 | 53.4 | 52.8 | 0.0 | 0.1 | 1.3 | 4.3 | 0.0 | 0.2 |
| - No Speech | 78.0 | 82.1 | 89.5 | 55.9 | 59.3 | 64.4 | 4.9 | 17.3 | 17.7 | 38.3 | 4.6 | 15.5 |
| - No Text | 78.5 | **83.2** | 89.0 | 54.8 | 58.3 | 63.4 | **6.3** | 17.4 | 20.3 | 42.3 | **7.0** | **16.3** |
| - No ASR & TTS | **78.7** | 81.4 | 89.5 | **56.7** | **59.5** | **68.5** | 6.1 | 17.0 | **23.0** | 41.0 | 5.4 | 14.0 |

(a) Sampling Rate

(b) Span Corruption Ratio

(c) Interleaved Data Tokens

Figure 3: (a) Sampling rate vs average accuracy. (b) Span corruption ratio vs average accuracy. The accuracy is averaged over datasets of speech language modeling and spoken question answering. (c) Interleaved data tokens vs average performance after supervised fine-tuning.

## 3.3 ABLATION STUDY

### 3.3.1 DATA SCALING AND COMPOSITION

Our pre-training corpus consists of text, speech data, speech-text interleaved data, and speech-text parallel data (from ASR and TTS tasks). We study the effects of data scaling and composition. First, we evaluate how scaling interleaved data impacts model performance. We train the 9B model with interleaved data sizes of 0, 100B, and 200B tokens, keeping other parts of the pre-training corpus unchanged. Table 5 compares these results with the best model trained on 600B interleaved data. Without interleaved data, the model performs poorly, but as interleaved data scales up, performance improves consistently. This demonstrates the effectiveness of scaling synthetic speech-text interleaved data. Figure 3c further shows that increasing interleaved data improves chatbot performance after supervised fine-tuning, both with and without text guidance. Next, we analyze the contributions of different parts of the pretraining corpus using a 1.5B model. Results in Table 5 show that removing synthetic interleaved data significantly degrades performance. Removing unsupervised speech data slightly reduces spoken question answering accuracy, while removing text or speech-text parallel data improves performance on most benchmarks, likely due to capacity competition among modalities in smaller models. For the 9B models, we retain all data types as they represent essential tasks for downstream applications, and larger models alleviate this competition.

### 3.3.2 FRAME RATE

The frame rate of the speech tokenizer refers to the number of speech tokens generated per second. Hassid et al. (2023) observed that reducing HuBERT's frame rate from 50Hz to 25Hz improved performance on speech language modeling tasks. We trained 1.5B models with tokenizers at different frame rates using the same number of training tokens, excluding ASR and TTS datasets for simplic-

ity, and analyzed the relationship between sampling rate and accuracy (Figure 3a). The results show that lower frame rates improve average accuracy. We hypothesize two reasons: (1) lower sampling rates allow the model to process more speech data within the same training token budget, and (2) shorter token sequences for the same audio reduce modeling difficulty. We selected a 12.5Hz frame rate for our main model, as the 6.25Hz tokenizer showed a trade-off where speech information loss outweighed accuracy gains.

### 3.3.3 SPAN CORRUPTION RATIO

The span corruption ratio decides the proportions of text and speech tokens in interleaved samples. With extreme corruption ratios close to 0 or 1 the interleaved samples are dominated by text or speech tokens. To study the effect of the ratio and determine the best value, we train multiple 1.5B models with interleaved data from different span corruption ratios and plot the results in Figure 3b. Overall, we find that the corruption ratios from 0.2 to 0.4 works well. Larger or smaller ratios result in a significant degradation of performance. Based on the results, we select 0.3 as the corruption ratio for our main model.

## 4 RELATED WORK

**Speech Tokenization** Speech tokenizers, which transform a audio clip into discrete tokens, can be categorized into two directions. The neural acoustic codecs (Zeghidour et al., 2022; Défossez et al., 2023; Kumar et al., 2023; Ji et al., 2024) target at reconstructing high-quality audio at low bitrates. The semantic tokens (Hsu et al., 2021; Chung et al., 2021) are extracted from speech representations learned with self-supervised learning on speech data. Speechtokenizer (Zhang et al., 2024) unifies semantic and acoustic tokens as different residual vector quantization (RVQ) layers, but it also suffers from expansion of sequence length. Cosyvoice (Du et al., 2024) proposes the supervised semantic tokenizer derived from a speech recognition model, and successfully apply the tokenizer to text-to-speech synthesis. The application of the tokenizer on speech language modeling is not explored.

**Speech-Text Pre-training** GSLM (Lakhotia et al., 2021) proposes the generative spoken language modeling task, which trains the next-token-prediction objective on discrete semantic tokens produced by self-supervised learning. AudioLM (Borsos et al., 2023) proposes a hybrid tokenization scheme that combines semantic tokens with acoustic tokens from a neural audio codec (Zeghidour et al., 2022). TWIST (Hassid et al., 2023) trains the speech language model using a warm-start from a pretrained text language model, specifically OPT (Zhang et al., 2022). Moshi (Défossez et al., 2024) scales up the size of natural speech data in TWIST to 7 million hours. Spirit-LM (Nguyen et al., 2024) further extends TWIST by adding speech-text interleaving data curated from speech-text parallel corpus. However, the scarcity of parallel corpus restricts the scale of interleaving data.

**End-to-End Spoken Chatbots** Early works in speech-to-speech models mainly focus on processing tasks like speech translation (Chen et al., 2021b; Ao et al., 2022). Since success of ChatGPT in text-based chatbots, many works have explored methods to develop speech-based chatbots that can understand and respond in speech. Speechgpt (Zhang et al., 2023) proposes to combine existing large language models (LLM) with discrete speech representations to obtain speech conversational abilities. Moshi (Défossez et al., 2024) proposes a full-duplex spoken dialogue framework based on their pretrained speech language model. Llama-omni (Fang et al., 2024) and Mini-omni (Xie & Wu, 2024) both propose light-weight alignment methods that transform an open language model into spoken chatbots.

## 5 CONCLUSION

This paper introduced a novel approach for scaling speech-text pre-training using supervised semantic tokens and synthetic interleaved data. By employing a supervised speech tokenizer and generating 600B tokens of interleaved data, we scaled our speech pre-training to 1 trillion tokens, achieving state-of-the-art performance in speech language modeling and spoken question answering tasks. We also developed an end-to-end spoken chatbot by fine-tuning our pre-trained model, demonstrating competitive performance in both conversational abilities and speech quality. Future work could explore more efficient training techniques, investigate larger model sizes, and expand multilingual capabilities.

## ETHICS STATEMENT

This research explores large-scale speech-text pre-training with interleaved speech-text data synthesized from text corpora, significantly improving speech modeling capabilities for SpeechLLMs. Furthermore, by fine-tuning the pre-trained model on speech dialogue data, we develop an end-to-end spoken chatbot with competitive performance in both conversational abilities and speech quality. While our method advances end-to-end spoken interaction, we analyze potential risks and provide possible ways to mitigate those issues.

**Voice Phishing** Voice Phishing is a deceptive practice where malicious actors use synthetic voices to imitate trusted individuals for fraudulent purposes. As speech synthesis technology advances, the risk of such abuse increases. In our approach, while the SpeechLLM generates speech tokens that encode semantic information, the speech decoder is responsible for synthesizing the actual speech waveform, with voice identity primarily determined by its training. The decoder's training follows a two-stage process: initial pre-training on multi-speaker data followed by fine-tuning on single-speaker data, resulting in a model that generates speech with one fixed voice identity(Cf. Appendix B.2). This design significantly reduces the potential for misuse in voice phishing, as the model cannot flexibly adapt to mimic multiple identities, a critical requirement for such fraudulent activities.

**Scamming** Beyond Voice Phishing, there exists a risk of Scamming, where the model could potentially be exploited to generate deceptive or fraudulent content for broader malicious purposes. Unlike traditional text-to-speech (TTS) models, which can generate speech from any text input, our model is an end-to-end spoken chatbot. This means it takes speech input from the user and produces speech output as a response, without the ability to generate speech from arbitrary text. This design greatly lowers the risk of misuse for scamming purposes, as the model is built for interactive speech-based conversations, not unrestricted audio generation. Additionally, our pre-trained model is fine-tuned on synthesized speech dialogues derived from ethically aligned LLMs (Cf. Section 2.3.2). This approach inherits the ethical constraints and safety measures already established in these LLMs. The inherited safeguards enable our model to exhibit similar ethical behavior patterns, such as refusing to engage in fraudulent or harmful content generation.

**Potential Risks of Fine-Tuning and Mitigation** Despite the safeguards embedded in our model's design, we recognize that fine-tuning could potentially undermine these protections. For instance, fine-tuning the speech decoder on diverse speaker data could enable the model to generate speech with multiple voice identities, thereby circumventing the single-voice limitation and increasing the risk of voice phishing. Similarly, fine-tuning the SpeechLLM on unaligned or malicious data could weaken its ethical alignment, allowing it to generate fraudulent or harmful content. To mitigate these risks, we advocate for strict access controls on fine-tuning capabilities, ensuring that only authorized users with ethical oversight can modify the model. Additionally, we recommend the development of robust detection mechanisms for synthetic speech, such as watermarking or forensic analysis tools, to identify and prevent misuse. Ethical guidelines should be rigorously enforced during any fine-tuning process, and transparency in model deployment should be prioritized to maintain accountability. By acknowledging these risks and implementing proactive measures, we aim to ensure that our SpeechLLM is used responsibly and ethically.

## ACKNOWLEDGEMENT

We would like to thank the anonymous reviewers for their suggestions in refining this work. This work is supported by NSFC 62425601, 62276148, and New Cornerstone Science Foundation through the XPLORER PRIZE, and a research fund from Tsinghua Guoqiang Institute.

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

# A  TRAINING DATASETS

## A.1  INTERLEAVED SPEECH-TEXT DATA

Table 6: Statistics on interleaved pre-training data. Tokens measured in billion.

| Dataset | Tokenizer | Corruption Ratio | Text Tokens | Speech Tokens | Speech Ratio | Total Tokens |
|---|---|---|---|---|---|---|
| Fineweb-Edu | Text-60k Speech-50Hz | 0.30 | 56.21 | 343.79 | 0.86 | 400 |
| | Text-60k Speech-25Hz | 0.30 | 98.78 | 301.22 | 0.75 | 400 |
| | Text-60k Speech-12.5Hz | 0.10 | 282.82 | 117.18 | 0.29 | 400 |
| | | 0.20 | 209.05 | 190.95 | 0.48 | 400 |
| | | 0.30 | 158.43 | 241.57 | 0.60 | 400 |
| | | 0.40 | 121.54 | 278.46 | 0.70 | 400 |
| | | 0.50 | 93.48 | 306.52 | 0.77 | 400 |
| | | 0.75 | 46.15 | 353.85 | 0.88 | 400 |
| | | 1.00 | 0.10 | 399.90 | 1.00 | 400 |
| | Text-60k Speech-6.25Hz | 0.30 | 226.50 | 173.50 | 0.43 | 400 |
| | Text-150k Speech-12.5Hz | 0.30 | 150.51 | 249.49 | 0.62 | 400 |
| Chinese-Fineweb-Edu | Text-60k Speech-12.5Hz | 0.30 | 78.80 | 121.20 | 0.61 | 200 |
| | Text-150k Speech-12.5Hz | 0.30 | 77.59 | 122.41 | 0.61 | 200 |

## A.2  SUPERVISED SPEECH DATA

Table 7: TTS training data of text-to-token LLM with 12.5Hz speech tokenizer. Tokens measured in billions.

| Language | Speech Hours | Speech Tokens | Text Tokens | Total Tokens |
|---|---|---|---|---|
| Chinese | 94,980 | 4.27B | 1.18B | 5.45B |
| English | 42,726 | 1.92B | 0.56B | 2.49B |

Table 8: Training data breakdown of ASR task for 12.5Hz speech tokenizer. Tokens measured in billion.

| Language | Speech Hours | Speech Tokens | Text Tokens | Total Tokens |
|---|---|---|---|---|
| Chinese | 21,624 | 0.97B | 0.42B | 1.39B |
| English | 68,733 | 3.09B | 1.06B | 4.16B |

## A.3 TEXT-TO-TOKEN MODEL DATA

Table 9: Training data of text-to-token LLM with 12.5Hz speech tokenizer synthesized by CosyVoice model on span-corruption data.

| Language | Speech Hours | Speech Tokens | Text Tokens | Total Tokens |
|---|---|---|---|---|
| Chinese | 9,895 | 0.45B | 0.13B | 0.58B |
| English | 11,074 | 0.50B | 0.16B | 0.65B |

The data used to train the text-to-token model consists of two parts: the TTS data presented in Table 7 and the synthesized data of incomplete text spans the generated by CosyVoice, as detailed in Table 9.

# B TRAINING DETAILS

## B.1 SPEECH TOKENIZER

We fine-tune the vector-quantized Whisper model with a collection of ASR datasets, including LibriSpeech (Panayotov et al., 2015), GigaSpeech (Chen et al., 2021a), MLS-Eng (Pratap et al., 2020), Wenet (Yao et al., 2021), CommonVoice (Ardila et al., 2020), AISHELL-1 (Bu et al., 2017), and a proprietary Chinese ASR dataset of 10k hours. We also include the unsupervised speech data with pseudo labels generated by Whisper (Radford et al., 2023) for English and FunASR (Gao et al., 2023) for Chinese.

All of our speech tokenizers in Table 1 are fine-tuned from `whisper-large-v3` for 2 epochs with batch size 4096 and learning rate 1e-5. The ratio of supervised samples to pseudo-labeled samples is 1:3. The codebook vectors are updated with exponential moving average with decay coefficient 0.99 and the commitment loss coefficient is 10.0. To reduce the information loss of average pooling, we increase the codebook size as the sampling rate decreases.

Table 10: **ASR results of Whisper models with pooling layers and vector quantization.** The LibriSpeech (English) is measured with word-error-rate (WER) and AISHELL-1 (Chinese) is measured with character-error-rate (CER). The first model is the original `whisper-large-v3` without fine-tuning.

| Model | Sampling Rate | VQ Codebook | Finetuned | LibriSpeech test-clean | test-other | AISHELL-1 test |
|---|---|---|---|---|---|---|
| whisper-large-v3 | 50Hz | - | No | 2.50 | 4.53 | 9.31 |
| whisper-large-v3 | 50Hz | 4,096 | Yes | 1.85 | 3.78 | 2.70 |
| whisper-large-v3 | 25Hz | 4,096 | Yes | 1.94 | 4.16 | 2.86 |
| whisper-large-v3 | 12.5Hz | 16,384 | Yes | 2.10 | 4.90 | 3.02 |
| whisper-large-v3 | 6.25Hz | 65,536 | Yes | 2.48 | 6.34 | 3.33 |

During training of speech tokenizers, we measure the semantic information with accuracy of automatic speech recognition (ASR) datasets. We evaluate the finetuned Whisper on LibriSpeech (Panayotov et al., 2015) and AISHELL-1 (Bu et al., 2017), along with the original Whisper model. The results are shown in Table 10. Overall all the tokenizers reserve enough semantic information to achieve accurate ASR performance.

Table 11: Ablation study on block sizes in the block causal attention of speech tokenizers.

| Model | Attention Type | Block Size | LibriSpeech test-clean | test-other | AISHELL-1 test |
|---|---|---|---|---|---|
| whisper-large-v3 | Bidirectional | - | 3.45 | 5.82 | 4.71 |
| whisper-large-v3 | Causal | - | 3.13 | 7.10 | 6.27 |
| whisper-large-v3 | Block Causal | 0.5s | 3.85 | 7.30 | 5.70 |
| whisper-large-v3 | Block Causal | 1s | 3.37 | 6.50 | 5.14 |
| whisper-large-v3 | Block Causal | 2s | 3.39 | 6.16 | 4.76 |

We also conduct an ablation study on the effect of block size in the block causal attention. In the ablation study we fine-tune the Whisper model with only the supervised ASR datasets for 20,000 steps with batch size 1024. For all the models we use VQ codebook size 4096 and sampling rate 25Hz. The results are shown in Table 11.

## B.2 Speech Decoder

The speech decoder uses the same architecture as the flow matching model in CosyVoice (Du et al., 2024). For simplicity, we exclude the speaker embedding component from the flow model. The decoder is trained from scratch for 2 epochs with dynamic batching of 20000 frames in a batch and learning rate 1e-3. The training datasets include Emilia (He et al., 2024), Yodas2 (Li et al., 2023a), Libri-Light (Kahn et al., 2020) and a proprietary Chinese speech dataset. To achieve consistent speaker identity in our end-to-end spoken chatbot, we further fine-tune the speech decoder using speech responses synthesized by MeloTTS in the SpeechDialog-90K dataset.

## B.3 Text-to-Token Model

The text-to-token model is initialized from a 1.5B pre-trained text LM of the same architecture (further experiments indicate that training from scratch yields the same performance). The training dataset consists of the TTS corpus in Table 7 and Table 9, with sampling ratio proportionate to the number of samples in each subset. We use the AdamW (Loshchilov & Hutter, 2019) optimizer with $\beta_1 = 0.9$ and $\beta_2 = 0.95$. The model is trained for 300B tokens with sequence length of 4096 and batch size of 256, learning rate that decays from $4 \times 10^{-4}$ to $4 \times 10^{-5}$, and weight decay 0.1. The architecture of the text-to-token model is shown in Table 12 (number of speech tokens not included in the vocab size).

Table 12: Model architecture of the text-to-token language model.

| Hyper-parameters | Value |
|---|---|
| Number of layers | 28 |
| Hidden size | 2048 |
| FFN inter hidden size | 6144 |
| Activation function | SwiGLU |
| Attention heads | 16 |
| Attention head size | 128 |
| Attention group size | 4 |
| Maximum sequence length | 8192 |
| Vocab size | 59264 |

## B.4 Speech Language Model

Table 13: Model architecture of the speech language model.

|  | 9B | 1.5B |
|---|---|---|
| Number of layers | 40 | 28 |
| Hidden size | 4096 | 2048 |
| FFN inter hidden size | 13696 | 6144 |
| Activation function | SwiGLU | SwiGLU |
| Attention heads | 32 | 16 |
| Attention head size | 128 | 128 |
| Attention group size | 2 | 4 |
| Maximum sequence length | 8192 | 8192 |
| Vocab size | 151552 | 59264 |

## C Evaluation Details

For Spoken StoryCloze and Spoken TopicStoryCloze, we synthesize the speech for contexts and continuations with the provided texts with the TTS engine. When selecting the most probable continuation, the likelihood is normalized by the number of tokens in each continuation.

For Llama Questions, we use the audio files provided in the dataset[1]. We synthesize the speech for Web Questions and TriviaQA with the TTS engine. For TriviaQA, we randomply sample 1,000 samples from the test set of the 'rc' setting to match the size of the other two datasets. For all the three datasets of spoken question answering, we add the text prompt "the answer is" after the question for both the 'S' and 'S→T' settings. For the 'S' setting the model generates speech of at most 10 seconds, and for the 'S→T' setting the model generates at most 128 tokens.

## D Case Study

### D.1 Spoken Question Answering

Here we provide examples of spoken question answering from Llama Questions, Web Questions, and TriviaQA. Given a question in speech, the speech language model predicts speech tokens, which are then fed into the speech decoder to get the output audio. We utilize *whisper-large-v3* to transcribe the audio into text.

Dataset: Llama Questions

```
Question: What is the capital of France? The answer is
Ground Truth: Paris
Prediction: Paris. The capital of France is Paris. Paris is the capital
of the, the, the, the,
```

Dataset: Web Questions

```
Question: Who is governor of ohio 2011? The answer is
Ground Truth: John Kasich
Prediction: John Kasich. What is the name of the governor? 2012. The
answer. John Kasich. What is
```

Dataset: TriviaQA

---

[1] https://github.com/google-research-datasets/LLAMA1-Test-Set

```
Question: Who plays the role of Bubble in Absolutely Fabulous? The
answer is
Ground Truth: Jane Horrocks
Prediction: Julia Steiner Julia Steiner is an American actor, singer and
 dancer. He S he S he
```

For the examples above we can observe that our model can answer questions about world knowledge.

## D.2 SPOKEN CHATBOT EVALUATION

Here we provide examples of spoken chatbot evaluation. The question and response are transcriptions of the speech input and output.

Task: General QA

```
Question: I need to get my nephew a birthday gift. I'm thinking of
giving him a game for his Nintendo Switch. Can you give me a few options
 that came out recently?
Response: What a great nephew you are. Giving a game for his Nintendo
Switch is a fantastic idea. Here are some amazing options that just
released. Super Mario Odyssey, The Legend of Zelda Tears of the Kingdom,
 Bookman Sword and Shield, Just Dance, Minecraft, The Elder Scrolls Five
 Skyrim Anniversary Edition, Portal Knights, Shovel Knight, Celeste, and
 Overcooked two. These games are highly rated and perfect for any
Nintendo Switch player. Choose one that matches his interests and enjoy
his birthday.
Score: 3
```

Task: Knowledge

```
Question: How many days does Hanukkah last?
Ground Truth: Eight
Response: Hanukkah, also known as the Festival of Lights, is an eight-
day Jewish holiday that commemorates the rededication of the Temple in
Jerusalem during the Maccabean Revolt. It typically falls in November or
 December, depending on the Hebrew lunar calendar, the festival is
observed for eight days and nights, with each day corresponding to one
of the eight brands of the menorah. Traditional activities include
lighting the menorah, being frayed foods like lakes and sufkat,
exchanging gift, playing dreidel and reciting the Hanukkah prayers. The
lighting of the menorah is a central ritual. With the first candle lit
on the earth's day, additional candles added each day. A festival
includes on the eighth day with the lighting of the final candle and the
 sabration of Shemini Atzeret and Simkat Torah. Some cultures celebrate
Hanukkah for Nidaz, including the Ishan al-Dev, Shemini Atzeret and
Simkat Torah.
Judgment: Correct
```

## D.3 INTERLEAVED PRE-TRAINING DATA

Here we provide examples of speech-text interleaved pre-training data. The following examples are sampled from FineWeb interleaved pre-training dataset with 150k text tokenizer and 12.5Hz speech tokenizer. The speech tokens are transcribed by the corresponding ASR model of the speech tokenizer and are displayed in blue and the special token in **bold**. An extra new line is added before and after the audio segment for clarity.

```
Eugene Van Reed
- Died: 1873
```

**<|begin_of_audio|>** [53 speech tokens] originally from san francisco, van reed first traveled to japan in **<|end_of_audio|>**
 1859, where he established his own trading company, dealing in arms
 among other goods. Named Consul General of
**<|begin_of_audio|>** [34 speech tokens] the hawaiian kingdom in eighteen sixty six **<|end_of_audio|>**
 he played a key role in
**<|begin_of_audio|>** [43 speech tokens] organizing for the first japanese immigrants to travel to hawaii in **<|end_of_audio|>**
 1868. This first group of 148, known as the gannenmono, encountered
 severely harsh working conditions on the sugar plantations, leading to
 considerable tension between the governments of Japan, Hawaii,
**<|begin_of_audio|>** [40 speech tokens] and the united states, resulting in official japanese, **<|end_of_audio|>**
 immigration to
**<|begin_of_audio|>** [71 speech tokens] hawaii at not beginning until 1885, after lengthy negotiations **<|end_of_audio|>**
 .
 Van Reed died in 187 3 , aboard a ship called Japan , which he had been
  taking home to San Francisco from Japan .

According to declarations from the sector Chambers, Argentina consumed
340 million litres of pesticides and herbicides in the last year; and
this quantity is increasing 15% to 20% each year. These poisons are
sprayed, fumigated and applied to areas inhabited by 12 million people.
For
**<|begin_of_audio|>** [20 speech tokens] a long time the residents **<|end_of_audio|>**
 of the affected localities have been denouncing to suffer from serious
 diseases as a consequence of their being contaminated by
**<|begin_of_audio|>** [52 speech tokens] the pestatites this situation was confirmed at the first **<|end_of_audio|>**
 and 2nd Meeting
**<|begin_of_audio|>** [73 speech tokens] of physicians of the fumigated towns, at the cordoba medical cns of faculty, and, **<|end_of_audio|>**
 Medical Sciences Faculty of the Rosario National University , in 2010
 and 2011, respectively.
There is
**<|begin_of_audio|>** [62 speech tokens] substantial public demand to reclassify pesticides in arginina.  this demand **<|end_of_audio|>**
 is sound: depending on how pesticides
**<|begin_of_audio|>** [77 speech tokens] are classified the provincial and municipal regulations determine the distances between fumigated **<|end_of_audio|>**
 (sprayed) and inhabited areas.
Currently, the classification is made according to the quantity in
milligrams of poison that, when fed to rats, kills 50% of the population
 tested (Lethal Dose test or LD50); the less the quantity of poisonous
substance is required,
**<|begin_of_audio|>** [34 speech tokens] the higher the level of toxicity is attributed **<|end_of_audio|>**
 to that substance. As such, this form of measurement ignores medium and
  long term effects, including oncogenic, reproductive
**<|begin_of_audio|>** [43 speech tokens] immunological and endocrine ones in the light of **<|end_of_audio|>**
 these facts,
**<|begin_of_audio|>** [38 speech tokens] glyphosate should be reclassified as level ib **<|end_of_audio|>**
 (highly hazardous; the WHO recommended classification of pesticides by
 hazard), particularly because of the scientific and epidemiological
 data, showing that its accumulation in the body is connected to
**<|begin_of_audio|>** [49 speech tokens] continental malformations and spontaneous abortions 1 **<|end_of_audio|>**

```
 3].
Furthermore, the current toxicological classification of acute effects
of pesticides
<|begin_of_audio|> [24 speech tokens] done taking to account
a <|end_of_audio|>
 new set of information and scientific data, which show the acute damage
  of these poisons for agricultural use on humans, and that are
 different from
<|begin_of_audio|> [38 speech tokens] the findings in rodents
highlighting patterns <|end_of_audio|>
 specific to humans.
```

## E    PROMPT FOR CONSTRUCTING SPEECH DIALOGUE DATASET

```
You are an AI assistant designed to convert text SFT data into SFT data
adjusted for speech synthesis tasks. Your task is to generate a modified
 response suitable for text-to-speech synthesis under the following
conditions:

- Exclusion of Unreadable Characters and Number Conversion: Remove any
characters that text-to-speech (TTS) systems cannot synthesize, such as
*, parentheses (), bullet points, or other special symbols. Convert all
numbers into their English word equivalents. For example, convert one to
 one, twenty to twenty, and so on. Do not include lists or line breaks;
the response should be a single paragraph.
- Specificity in Response: Make the response more specific and to the
point, avoiding lengthy explanations. Focus on delivering the key
message concisely.
- Clarity: Ensure that the response is clear and easy to understand when
 spoken aloud.
- Avoidance of Code Content: If the prompt suggests writing or
generating code, return an empty JSON object. If the prompt only
inquires about knowledge related to code without requiring code
generation, provide a modified response.

Below is the the conversation input:

[Prompt]: {prompt}
[Response]: {response}

Please output in the following JSON format if the conditions are met:

```json
{"response": "<modified_response>"}
```

If the prompt is filtered out, output: {}
```

## F    PROMPT TEMPLATE FOR SPOKEN DIALOGUE

**Direct Generation**

```
<|system|>
User will provide you with a speech instruction. Think about the
instruction and speak the response aloud directly.
<|user|>
<|begin_of_audio|>{Speech Instruction}<|end_of_audio|>
<|assistant|>
<|begin_of_audio|>{Speech Response}<|end_of_audio|>
```

**Text-guided Generation**

```
<|system|>
User will provide you with a speech instruction. Think about the
instruction and speak the response aloud directly.
<|user|>
<|begin_of_audio|>{Speech Instruction}<|end_of_audio|>
<|assistant|>transcript
{Text Response}
<|assistant|>
<|begin_of_audio|>{Speech Response}<|end_of_audio|>
```

# G   PROMPT FOR EVALUATING SPOKEN CHATBOTS

**General QA**

```
[Instruction]
Please act as an impartial judge and evaluate the quality of the
response provided by an AI assistant to the user question displayed
below. Your evaluation should consider factors such as the helpfulness,
relevance, accuracy, depth, creativity, and level of detail of the
response. Begin your evaluation by providing a short explanation. Be as
objective as possible. After providing your explanation, you must rate
the response on a scale of 1 to 10 by strictly following this format:
"[[rating]]", for example: "Rating: [[5]]".

[Question]
{instruction}

[The Start of Assistant's Answer]
{response}
[The End of Assistant's Answer]
```

**Knowledge**

```
Your will be given a question, the reference answers to that question,
and an answer to be judged. Your tasks is to judge whether the answer to
 be judged is correct, given the question and reference answers. An
answer considered correct expresses or contains the same meaning as at
least **one of** the reference answers. The format and the tone of the
response does not matter.

You should respond in JSON format. First provide a one-sentence concise
analysis for the judgement in field 'analysis', then your judgment in
field 'judgment'. For example,
```json
{{"analysis": <a one-sentence concise analysis for the judgement>, "
judgment": <your final judgment, "correct" or "incorrect">}}
```

# Question
{instruction}

# Reference Answer
{targets}

# Answer To Be Judged
{answer_to_be_judged}
```

