# OpenReview forum: "Scaling Speech-Text Pre-training with Synthetic Interleaved Data"
_ICLR.cc/2025/Conference — ICLR 2025 Poster_

### Official Review · Reviewer_LiYj · 2024-10-25

**Soundness:** 3
**Presentation:** 3
**Contribution:** 4
**Rating:** 8
**Confidence:** 4

**Summary:**

This work  present a novel approach for scaling speech-text pre-training by leveraging large-scale synthetic interleaved data derived from existing high-quality text corpora. This work utilized existing text-to-speech (TTS) datasets to train a text-to-token language model, which is used to synthesize 600B tokens of speech-text interleaved data. Experiments have demonstrated the effectiveness of incorporating interleaved speech-text data, which can effectively align speech and text. Furthermore, this work constructs a speech instruction dataset, SpeechDialog-90K, to fine-tune models into a chatbot model, which can directly generate speech responses without intermediate text response and significantly improve the previous SOTA.

**Strengths:**

1. This work demonstrates the need to use interleaved speech-text data for cross-modal pre-training, paving the way for more effective speech-text pretraining. Previous pretraining methods have typically relied on paired ASR or TTS data, which is limited in scale; or independently utilized unsupervised speech and text data, which cannot model the dependency between the two modalities. I believe that this work makes a great contribution to the filed of speech-text pretraining.

2. Although the use of interleaved data has been proven effective in the field of image-text pretraining [1], it has not been explored in the field of speech-text pretraining. In contrast to the vision field, web data can naturally form interleaved image-text data, but it is difficult to collect real data with interleaved speech and text. This work proposes a novel method to synthesize pseudo-interleaved data using a text-to-tokens model, and through thorough experimentation, demonstrates the effectiveness of synthesized data and observes that scaling synthesized data continues to provide benefits.

3. This work is very solid and well-motivated. The paper is well-structured, with a clear presentation of the methodology, experiments, and results. This work also reports state-of-the-art performance in speech language modeling and spoken question answering.

[1] Chameleon team. Chameleon: Mixed-Modal Early-Fusion Foundation Models. arxiv: 2405.09818.

**Weaknesses:**

1. There is a lack of performance evaluation for the text-to-tokens model. For example, after converting a piece of text into tokens and then decoding it into speech using a vocoder, what is the ASR-WER of the resulting speech? This result is necessary to demonstrate the semantic representation capability of the tokens generated by the text-to-tokens model.

2. Based on my experience, tokens generated by text-to-models lacks diversity, and the speech instruction dataset SpeechDialog-90K in the SFT stage is also synthesized by TTS, so I am concerned about whether the model can understand real speech input. I checked the evaluation datasets in this work, all of which were synthesized through TTS, lacking evaluation on real speech input (such as AIRBench [2]).

3. The quality of the output speech is not satisfactory, as evidenced by the poor ASR-WER in Table 4. In comparison to llama-omini, which was only trained on 100 hours of speech data, this model was trained on a much larger scale of 700k hours of speech data. The author needs to provide a reasonable explanation for why the ASR-WER is so poor.

[2] Yang, Qian, et al. AIR-Bench: Benchmarking Large Audio-Language Models via Generative Comprehension. arxiv 2402.07729.

**Questions:**

1. I'm curious if there is any additional filtering process for the text input into the text-to-tokens model, as there are many texts that cannot be synthesized, such as code or mathematical formulas.

2. Why is it required for the encoder and decoder to be causal when training the speech tokenizer? During the inference stage, speech are segmented into 2s-chunks for inference, which does not require a streaming speech tokenizer.

---

> ### Author Response · Authors · 2024-11-22
> **Response to Reviewer LiYj**
>
> Thank you for your review. We are pleased to hear that you find this work solid and well-motivated. Please let us know if your concerns and questions have been addressed—we would be happy to engage in further discussions!
>
> > There is a lack of performance evaluation for the text-to-tokens model. For example, after converting a piece of text into tokens and then decoding it into speech using a vocoder, what is the ASR-WER of the resulting speech? This result is necessary to demonstrate the semantic representation capability of the tokens generated by the text-to-tokens model.
>
> We have added more details about the evaluation of text-to-token model in Section 2.2. Table 2 shows the ASR-WER of the results speech. For the standard VCTK dataset, we observed a lower WER of 3.20. However, the WER for speech spans generated from the text pre-training data was higher. We attribute this discrepancy primarily to the format and content of the text pre-training data, which can include text that is challenging to pronounce accurately.
>
> > Based on my experience, tokens generated by text-to-models lacks diversity, and the speech instruction dataset SpeechDialog-90K in the SFT stage is also synthesized by TTS, so I am concerned about whether the model can understand real speech input. I checked the evaluation datasets in this work, all of which were synthesized through TTS, lacking evaluation on real speech input (such as AIRBench [2]).
>
> Thanks for point out this! We explain this in two aspects:
>
> - First, we have conducted additional evaluations on ASR benchmarks, specifically LibriSpeech for English and AISHELL-1 for Chinese, using our base model. The results demonstrate competitive performance, indicating that our architecture is fully capable of understanding real speech input.
>
>     | Method | LibriSpeech (test-clean) | LibriSpeech (test-other) | AISHELL-1 (test) |
>     | ---- | ---- | ---- | ---- |
>     | whipser-large-v3 | 2.50 | 4.53 | 9.31 |
>     | 9B-base (ours) | 2.82 | 7.66 | 2.46 |
>
> - Second, we were unable to find open-source speech dialogue data with real and diverse speech input. Therefore, to demonstrate the model's potential as an end-to-end spoken chatbot, we synthesized speech dialogue data using TTS model. Regarding benchmarks, while AIRBench focuses more on audio understanding tasks (e.g., speaker gender recognition, music classification), we believe it may not be suitable for evaluating a spoken chatbot system. Since the primary focus of this work is on pre-training, we synthesized speech through TTS for evaluation purposes. However, we are committed to exploring new datasets and benchmarks in the future to further evaluate and enhance our model.
>
> > The quality of the output speech is not satisfactory, as evidenced by the poor ASR-WER in Table 4. In comparison to llama-omini, which was only trained on 100 hours of speech data, this model was trained on a much larger scale of 700k hours of speech data. The author needs to provide a reasonable explanation for why the ASR-WER is so poor.
>
> We apologies for this. The poor ASR-WER was caused by an implementation mistake in the speech reconstruction process. This issue resulted in some speech tokens being randomly omitted, leading to mispronounced words. It is important to note that this mistake only affected the spoken chatbot results. We have corrected the error and updated Table 4 with the accurate evaluation results. The updated results show that our model achieves a lower WER of 7.83 compared to LLaMA-Omni's 7.95. We believe the WER could be further reduced with the use of a more advanced TTS model for speech dialogue data construction.

---

> ### Comment · Reviewer_LiYj · 2024-11-26
>
> Thank the author for the detailed reply.
>
> I have seen the evaluation of the text-to-tokens model, the evaluation on the ASR task, and the corrected results in Table 4 in the new version. I'm glad to see that the final end-to-end model can achieve a relatively low WER on the ASR task. However, it can also be seen that the text-to-tokens model has a relatively poor semantic modeling ability on real text corpora.
>
> Nevertheless, I'm still quite curious about why the author designed the speech tokenizer to be casual. During inference, speech is segmented into chunks of 2 seconds before being fed into the LLM, rather than being in a completely streaming manner.
>
> Despite some limitations of this article, I'm still inclined to accept it.

---

> > ### Author Response · Authors · 2024-11-27
> >
> > Thank you for your feedback! As shown in Table 11, we conducted an ablation study on block size variations (full causal, 0.5s, 1s, 2s) and found that the 2-second block configuration closely matches the ASR performance of bidirectional attention. The primary reasons for adopting causality are: (1) enabling streaming encoding for long audio inputs to reduce latency and (2) aligning with the autoregressive nature of SpeechLM. We find that the 2-second block size strikes a good balance between latency and performance. Accordingly, the speech tokenizer we used (see Section 2.1) was trained with a 2-second block causal approach.

---

### Official Review · Reviewer_XoEC · 2024-10-26

**Soundness:** 2
**Presentation:** 1
**Contribution:** 3
**Rating:** 5
**Confidence:** 4

**Summary:**

This paper proposes a method for scaling SpeechLMs using a new pre-training scheme called "Synthetic Interleaved Data". In this scheme, a text to token lM is first trained on supervised data, and then used to expand text-based datasets by predicting the speech tokens directly from the text data. For pre-training, only spans of text are converted to speech tokenization and text and speech are interleaved with one another. After this pre-training, the model is trained in the usual SpeechLM format. The strength of this approach is the ability to generate a large-scale dataset from text-corpora. Furthermore, this method shows strong results on speech-understanding and generation datasets.

Overall, the method presented in this paper and novel, and has an interesting contribution. On the other hand, the writing is unclear and the evaluations are somewhat lacking. I thus recommend to borderline reject this paper.

**Strengths:**

1. A novel approach for expanding the training corpora of SpeechLMs.
2. The method is easy to implement, and thus can be expanded to other methods.
3. State of the art results on sTopic-StoryCloze, sStoryCloze, Web Questions, Llama Questions and TriviaQA.

**Weaknesses:**

1. The writing is unclear for most parts of the paper. While Synthetic Interleaved Data is the main contribution of the paper, it is not clear what this means from the abstracts and introduction. Furthermore, the main explanation of this is just a small part of the paper. I would suggest reducing the length and condensing the section regarding speech tokenization (as this is a well established concept) and increasing the amount of detail in the section regarding Synthetic Interleaved Data. I would also suggest adding a better summerization of this concept in the introduction.
2. While there is a good ablation analysis, there aren't any explanations for why the architectural / training parameters where chosen as they where. I suggest to add a dedicated subsection or add this into the methodology section where the parameters are introducted.
3. The training datasets are lacking in clarity:
- For table 1, is it unclear on which dataset it was evaluated and why MOSNet scores are so low.
- It is unclear what Supervised speech-text data are used to train the model.
- It is unclear what datasets areused to train the text to speech tokens model.
- It is unclear what datasets are used to fine-tune the tokenization encoder and decoder.
These should be added in the section specifying the training pipeline or in a dedicated table / figure.
4. The experimental results are lacking in clarity:
- the origin of the baseline numbers in all tables is lacking, are these from other papers or from independently evaluating? I would suggest adding these directly to the table or in the caption.
- In table 3, speechGPT and Spectron are speech to speech methods, while the results are stated in speech to text.
- in Table 1 MOSNet was used while in table 4  UTMOS is used. This reason for this should be explained in the paper or have uniformity between them.
5. The paper is lacking some evaluations:
- Human evaluations of speech quality, such as MOS or MUSHRA evaluations where humans will rate the speech quality of the proposed method compared to the baseline.
- Evaluation on other tasks, such as speech continuation, reconstruction and TTS for the full method. Speech continuation and Reconstruction results at least should be added, while TTS might be left for future work.

**Questions:**

1  Why is the method pre-trained on Chinese data?
2. Why is GPT-4 used for scoring?
3. Why is whisper used as the encoder? did it perform better than other encoders?
4. It is stated that the model can do streaming, was this evaluated?
5. Why are the ablation also done on a 1.5B model?

---

> ### Author Response · Authors · 2024-11-22
> **Response to Reviewer XoEC (1/2)**
>
> Thank you for highlighting the weaknesses in our work and providing valuable suggestions to improve our writing and presentation. Based on your feedback, we have updated the paper and made several improvements, including adding more details about our method to enhance clarity and understanding.
> Specifically, we have added more details about the interleaved data construction process (Cf. Section 2.2) and better established this concept in the abstract and introduction. Additionally, we have included detailed training data statistics (Cf. Appendix A) and provided comprehensive training details for our model (Cf. Appendix B). We hope these changes better clarify our method and address your concerns. If they do, we would greatly appreciate it if you would consider raising your score. Thank you once again for your valuable feedback!
>
> > While there is a good ablation analysis, there aren't any explanations for why the architectural / training parameters where chosen as they where. I suggest to add a dedicated subsection or add this into the methodology section where the parameters are introducted.
>
> We have add explantations into method section, including sampling rates (Cf. Section 2.1), span corruption ratio (Cf. Section 2.2) and the amount of synthetic interleaved data tokens (Cf. Section 2.2).
>
> > The training datasets are lacking in clarity:
> For table 1, is it unclear on which dataset it was evaluated and why MOSNet scores are so low.
> It is unclear what Supervised speech-text data are used to train the model.
> It is unclear what datasets areused to train the text to speech tokens model.
> It is unclear what datasets are used to fine-tune the tokenization encoder and decoder. These should be added in the section specifying the training pipeline or in a dedicated table / figure.
>
> We haved add more details about the training dataset in Appendix A. We list the source and during of supervised speech-text data (including ASR and TTS data) in Appendix A.2. We list the dataset use to train text-to-token LM in Appendix A.3. We also add a section about the the training details of the speech tokenizer, speech decoder and speech language model in Appendix B.
>
> > The experimental results are lacking in clarity:
> the origin of the baseline numbers in all tables is lacking, are these from other papers or from independently evaluating? I would suggest adding these directly to the table or in the caption.
>
> We have added the source of baseline numbers to the caption of Table 1, Table 3 and Table 4. Thanks for your suggestion!
>
> > In table 3, speechGPT and Spectron are speech to speech methods, while the results are stated in speech to text.
> in Table 1 MOSNet was used while in table 4 UTMOS is used. This reason for this should be explained in the paper or have uniformity between them.
>
> The reason we choose different metrics is because we follow the settings of previous work for speech reconstruction and spoken chatbot. For the speech reconstruction, we adopt the approach from Moshi [1], utilizing the MOSNet metric. For the spoken chatbot evaluation, we follow the settings from Llama-Omni [2], using UTMOS as the evaluation metric. We haved add explantion to the following section.
>
> [1] Défossez, Alexandre, et al. "Moshi: a speech-text foundation model for real-time dialogue." arXiv preprint arXiv:2410.00037 (2024).
>
> [2] Fang, Qingkai, et al. "Llama-omni: Seamless speech interaction with large language models." arXiv preprint arXiv:2409.06666 (2024).
>
> > Human evaluations of speech quality, such as MOS or MUSHRA evaluations where humans will rate the speech quality of the proposed method compared to the baseline.
>
> Thank you for your suggestion. As the focus of this work is on pre-training, we fine-tuned the pre-trained model only on synthesized speech dialogue data, with the output speech generated using an open-source TTS system (MeloTTS in our experiments). This was intended to demonstrate the model's potential as an end-to-end spoken chatbot. Therefore, we did not conduct human evaluations of speech quality. We will consider adding human evaluations in future revisions.

---

> > ### Author Response · Authors · 2024-11-22
> > **Response to Reviewer XoEC (2/2)**
> >
> > > Evaluation on other tasks, such as speech continuation, reconstruction and TTS for the full method. Speech continuation and Reconstruction results at least should be added, while TTS might be left for future work.
> >
> > We evaluate speech continuation following the setting of Spectron, measuring the semantic quality with perplexity of GPT-2 medium.
> >
> > | Method | Perplexity |
> > | ---- | ---- |
> > | GSLM | 296.99 |
> > | AudioLM (3-RVQ) | 138.96 |
> > | AudioLM (12-RVQ) | 140.28 |
> > | TWIST (1.3B) | 229.53 |
> > | TWIST (7B) | 170.81 |
> > | SpeechGPT | 136.42 |
> > | Spectron (350M) | 126.08 |
> > | Ours (9B) | 30.21 |
> >
> > However, we think perplexity of GPT-2 medium is not a reliable metric due to the limitation of pretrained language models. We think the speech language modeling task evaluated in our paper are more suitable to evaluate speech continuation. The model is required to select the correct continuation from multiple plausible ones. It provides more accurate results than open-form generation.
> >
> > We evaluate speech reconstruction for our speech tokenizer and speech decoder in Section 2.1. We also evaluate our model for the TTS task.
> >
> > | Method | LibriTTS (test-clean) | SeedTTS (test-en) |
> > | ---- | ---- | ---- |
> > | Ours (9B) | 5.64 | 2.91 |
> >
> > > Why is GPT-4 used for scoring?
> >
> > We mainly follow MT-Bench [3] which use GPT-4 for scoring LLM outputs. With carefully designed prompts, we find GPT-4 provides reasonable and reliable evaluations.
> >
> > [3] Zheng, Lianmin, et al. "Judging llm-as-a-judge with mt-bench and chatbot arena." Advances in Neural Information Processing Systems 36 (2023): 46595-46623.
> >
> > > Why is whisper used as the encoder? did it perform better than other encoders?
> >
> > Whisper is a speech recognition model trained on a large and diverse audio dataset, demonstrating strong performance on downstream ASR tasks. Previous audio understanding models (e.g., Qwen-Audio [4]) have also utilized the Whisper encoder as an audio encoder, achieving excellent results. Therefore, we also adopt Whisper as our encoder and further train it with a VQ bottleneck to serve as a discrete speech tokenizer.
> >
> > [4] Chu, Yunfei, et al. "Qwen-audio: Advancing universal audio understanding via unified large-scale audio-language models." arXiv preprint arXiv:2311.07919 (2023).
> >
> > > It is stated that the model can do streaming, was this evaluated?
> >
> > We train the speech tokenizer and decoder to support streaming by processing audio in 2-second blocks, enabling speech output to be generated before all audio tokens are produced (25 speech tokens in a 12.5 Hz tokenizer). An ablation study on different block lengths of the speech tokenizer is presented in Table 11 (Cf. Appendix B.1). However, a detailed analysis of end-to-end latency of is left for future work, as this study primarily focuses on the pre-training phase.
> >
> > > Why are the ablation also done on a 1.5B model?
> >
> > We conducted the ablation study on the amount of interleaved data using the 9B model. Other ablations, such as sampling rate and span corruption ratio, were performed on the 1.5B model due to resource constraints, as training with the full interleaved dataset would require significantly more resources.

---

> > > ### Comment · Reviewer_XoEC · 2024-11-26
> > >
> > > Thanks a lot for the detailed response.
> > > The response satisfies almost all of the concerns in a good fashion, I thus raised the score.

---

> > > > ### Author Response · Authors · 2024-11-27
> > > >
> > > > Thank you for your kind feedback and for letting us know you’re satisfied. We noticed that the score hasn’t been updated yet—please let us know if there’s anything further we can clarify. We’d be more than happy to discuss this further!

---

### Official Review · Reviewer_kEHP · 2024-10-30

**Soundness:** 3
**Presentation:** 4
**Contribution:** 4
**Rating:** 8
**Confidence:** 4

**Summary:**

This paper tackles the important problem of closing the gap between textLMs and SpeechLMs, to enable a spoken conversation with an AI model.
The paper leverages the recently proposed *supervised* semantic tokens (introducing a discrete bottleneck in ASR models) which show better alignment with text.
Moreover, they train a text to audio tokens to enable the generation of synthetic audio tokens based on high-quality texts.
They suggest randomly replacing textual tokens with the corresponding synthetic speech tokens (resulting in an interleaved sequence), which helps to align the text and audio tokens.
They train large SpeechLMs on diverse text/audio inputs (audio only, text only, interleaved, [text,audio] and [audio,text]), and show convincing results on SLM, SQA.
They perform supervised finetuning on a proprietary (?) spoken dialogue dataset, and evaluate their model as a spoken chatbot using GPT-4 as a judge.

**Strengths:**

- Supervised speech tokenizers are a great way to distill the content from audio.  Audio is high-dimensional, and using text and a low bitrate bottleneck to focus on content is a good idea, suitable for SpeechLMs.
- Training a “TTS” model to generate synthetic audio tokens is interesting - as it doesn’t require generating the final audio (high-bitrate, compute-intensive, issues with OOD synthetic data).  Instead, they generate latent audio tokens that focus on content.
- the interleaving (replacing spans of text with its synthetically produced speech tokens) is interesting as it forces the model to learn alignment between text and audio tokens. It was also shown to be effective in practice.
- The ability to perform text-guided response (which is a kind of chain-of-thought) is interesting.
- The ablation study was done well.

**Weaknesses:**

- Several methodological evaluation details are missing (what was measured and how was it computed), mostly in Section 2.1 and Table 1 (See questions).  Whenever you report some metric with an intuitive non-exact name (e.g., Content Preservation - LS), you should explain somewhere it more precisely (e.g., Content Preservation: We run our quantized whisper on the LS (LibriSpeech) dataset to generate text and report the WER to the GT transcript) I understand that there’s a space limitation, but this is important.
I've listed some specific details I found missing in the Questions section. I suggest adding a short sub-paragraph that describes the evaluation methodology (defines all datasets+metrics being used) or adding those details into the main text within the relevant sections. If space is an issue, you can add those into an appendix section.

- Currently there's no sample page (unless I missed something). Consider creating a sample page with samples on speech continuation (audio prefix, audio GT continuation, and the model's audio continuation). Also consider adding examples of spoken question answering (audio question, audio GT answer, the model's prediction). Examples from the spoken chatbot evaluation would also be great. Moreover, you can visualize how the interleaved samples sound like (paragraph with audio tokens that were decoded in it).

**Questions:**

Please consider clarifying the following questions in the final revision, (apologies if I missed some details that are in the paper):

Section 2.1 and table 1:
- I’m assuming that you measured WER in content/semantic retention - is that right?
- How did you measure the ground truth (Whisper?)?
- The title says ‘Speech Tokenizer Results” but the Quality also measures your speech decoder.
- Is the speech decoder single/multi-speaker? Please add information regarding speaker identity preservation.
- What was the training data of the speech decoder? (single or multi speaker?)
- Missing citations (e.g. Expresso).
- LS stands for Librispeech? this isn’t stated.

Table 2:
- How do you measure WER? (those tokens into the bottleneck layer of the quantized Whisper? Decode the audio using the speech decoder and apply the regular Whisper?)

Section 2.2:
- Regarding \eta, please clarify if that is the ratio of text that is replaced or the ratio that audio will take out of the final sequence. I assume it is the first based on section A.1 but better for it to be clear in the main text too.

Section 2.3.2
In the main text, please add that you used GPT-4 to filter examples, shorten the responses, and avoid outputting text that cannot be read aloud (help the reader understand it from the main text). Also consider adding optimization details on the finetuning step (lr, batch size, etc')


Section 3.1:
- Please help the reader and add in the main paper that the GPT-4 content quality is on a scale of 1-10.
- The response of the SpeechLM was converted to text before it moved to GPT-4, right? How was the conversion performed? (Quantized whisper? your SpeechLM? Decoded and then used regular whisper?)
- Two comments regarding the ASR-WER metric in Table 4. First, it is more of a content quality metric rather than a speech quality metric. Secondly, as there are many ways to answer a question correctly, I suggest moving to ASR-ROUGE or ASR-BLUE instead.

Table 3:
- what’s the difference between \emptyset and “-“?


Comments:

- Fig1a is hard to understand at first glance - specifically, that the yellow tokens are replaced with speech tokens. Adding a color legend (Yellow: SpeechTokens, Cyan: TextTokens) would make it easier to understand.

- The second row in Fig 2a needs to be clarified. “Text—Audio token—>Text-to-token LM” can be perhaps: “Text — (TexttoTokenLM)—> Audio Tokens”.
- Semantic tokens determine the pacing of speech, which is a part of the speaker’s prosody. Your synthetic audio tokens are not conditioned on a speaker, so you are likely to get the semantic tokens of an average speaker. It is fine overall.

- Regarding audio tokenization - consider reducing the dimension (e.g. from 1024 to 8/16) before quantization to prevent codebook collapse.
- Moreover, I suggest applying text-tokenization algorithms (BPE?) on the speech units, to produce variable-length representations with a more balanced distribution, and further shorten the audio sequence.

Typos:
- Line 514: AudioLLM->AudioLM
- Line 516: Moshi Citet->citep

---

> ### Author Response · Authors · 2024-11-22
> **Response to Reviewer kEHP**
>
> Thank you for your detailed review and for providing many practical suggestions on improving the paper's presentation! We have made several changes to the paper, and we hope these updates better clarify our method and address your concerns. We would also be happy to engage in further discussions!
>
> > Currently there's no sample page (unless I missed something). Consider creating a sample page with samples on speech continuation (audio prefix, audio GT continuation, and the model's audio continuation). Also consider adding examples of spoken question answering (audio question, audio GT answer, the model's prediction). Examples from the spoken chatbot evaluation would also be great. Moreover, you can visualize how the interleaved samples sound like (paragraph with audio tokens that were decoded in it).
>
> - In Section 2.1 and Table 1, we provide additional details about the evaluation process, including the dataset name. Specifically, "LS" refers to LibriSpeech. To evaluate semantic retention, we measure the Word Error Rate (WER) using the ASR model provided in Expresso [1]. The speech decoder is trained on a multi-speaker TTS dataset (see Appendix A.2). Since the tokenizer is trained in a supervised manner, it effectively retains semantic content but does not preserve speaker identity.
> - In Section 2.2, we reorganized the section and included more details to describe the interleaved data construction process.
> - In Section 2.3.2, we added more details about the filtering process, including training hyperparameters.
> - In Section 3.1, we expanded on the spoken chatbot evaluation, specifying that we use whisper-large-v3 to transcribe generated speech into text for assessment.
> - We have included sample pages for spoken question answering, the spoken chatbot, and speech-text interleaved data. See Appendix E for more details.
>
> [1] Nguyen, Tu Anh, et al. "Expresso: A benchmark and analysis of discrete expressive speech resynthesis." arXiv preprint arXiv:2308.05725 (2023).
>
> > what’s the difference between \emptyset and “-“?
>
> We use $\emptyset$ to indicate tasks and modalities not supported by the model, and - to indicate scores that are not publicly available. We have add this explantion to the caption of Table 5.
>
> > Regarding audio tokenization - consider reducing the dimension (e.g. from 1024 to 8/16) before quantization to prevent codebook collapse.
>
> We indeed observed codebook collapse in our early experiments, we solve this issue by applying the random restart trick (Cf. Sec 2.1). We will try your solution in our following experiments.
>
> > Moreover, I suggest applying text-tokenization algorithms (BPE?) on the speech units, to produce variable-length representations with a more balanced distribution, and further shorten the audio sequence.
>
> We tried applying BPE on the speech units in our early experiments, but the compression ratio was not ideal. We learned a BPE tokenizer with 16k vocabulary on the 25Hz speech tokenizer with 4k vocabulary. The compression ratio (sequence length before BPE vs after BPE) is only 1.23, much smaller than BPE tokenizers for texts. Therefore we choose to reduce the sampling rate of speech tokenizer from 25Hz to 12.5Hz to reduce the sequence length.

---

> > ### Comment · Reviewer_kEHP · 2024-11-23
> >
> > Thank you for your detailed reply.
> >
> > > We have included sample pages for spoken question answering, the spoken chatbot, and speech-text interleaved data. See Appendix E for more details.
> > I still can't see the sample page that was included. Appendix E contains "PROMPT FOR CONSTRUCTING SPEECH DIALOGUE DATASET".
> > I expected to find a link to an anonymized webpage that contains audio examples (GT audio, Vocoder Resynthesis, Model prediction).
> > > Consider creating a sample page with samples on speech continuation (audio prefix, audio GT continuation, and the model's audio continuation). Also consider adding examples of spoken question answering (audio question, audio GT answer, the model's prediction). Examples from the spoken chatbot evaluation would also be great
> > Sharing an anonymized webpage with audio samples (which doesn't have to look impressive or contain many samples) would be valuable to the review process.
> >
> > I did see a visualization of the interleaving (in text form) in Appendix D.3 - thank you.

---

### Official Review · Reviewer_LeY9 · 2024-10-31

**Soundness:** 3
**Presentation:** 3
**Contribution:** 3
**Rating:** 8
**Confidence:** 3

**Summary:**

This paper is about scaling up data to train large speech language models.  The authors present a method for tokenizing speech using the Whisper encoder and demonstrate their tokenizer retains semantic information as well is fine-grained information for good quality speech generation.  They also describe a method for training a text-to-token model.  With these, they are able to tap into large resources of text data to generate synthetic training data, which they interleave with other conventional text and speech/text sources to pre-train a speech LM.  By fine-tuning the LM on a dialogue corpus they demonstrate a speech chat-like capability.  Extensive experimentation is performed, and the speech pretrain method does quite well on a range of tasks.

**Strengths:**

This paper is a nice contribution to the very hot topic of speech LMs.  By developing an effective speech tokenizer and text-to-tokenizer model the authors are able to create a very large speech language model that produces impressive results on a wide range of tasks.  The authors perform extensive experiments and ablation studies on the speech tokenizer, speech generator (decoder), and the speech LM.  The model is able to achieve strong performance on both spoken language modeling and spoken question answering tasks.  Finally, when fine-tuned on dialogue data, the model does well on a spoken chat-bot task.

**Weaknesses:**

Although this is not necessarily a weakness, this paper seems very strong on the engineering side and a little weaker on the novelty side of things.  The recipe the authors put forward consists of three separate steps 1) tokenizer, 2) text-to-token model 3) pretrain speech LM.   While the authors build a strong tokenizer based on the Whisper model, the approach is not especially novel as it is built on top of a strong speech recognition model.  Likewise the use of a TTS corpus to learn a text-to-token model is a nice approach, but has been done before to learn similar kinds of models (e.g., Hsu et al., Text-Free Image-to-Speech Synthesis Using Learned Segmental Units, 2020).  Finally, the interleaving of different kinds of text and speech data to pretrain an LLM with an additional token vocabulary is not especially novel.  However, while these points are arguably true, I find it impressive that the authors have put all the pieces together to create a very strong speech LM.

**Questions:**

While the whisper ASR model has achieved excellent performance on a range of tasks, it does have its limitations, especially with regards to unseen or low-resource languages.  That is not an issue for this paper which seems to focus on English (although there was quite a bit of Chinese data used as well).  Have the authors given any thought as to how to extend this work to cover more languages?

Are there any plans to open source the tokenizer, text-to-token model, or the speech LM itself?

Also, it would be nice if the authors could describe the amount of computation required to pretrain the speech LM.

---

> ### Author Response · Authors · 2024-11-22
> **Response to Reviewer LeY9**
>
> Thank you for your review and the positive assessment of our paper. We are pleased that you recognize our contributions to speech language models. If you have any further questions or suggestions, please don't hesitate to let us know!
>
> > While the whisper ASR model has achieved excellent performance on a range of tasks, it does have its limitations, especially with regards to unseen or low-resource languages. That is not an issue for this paper which seems to focus on English (although there was quite a bit of Chinese data used as well). Have the authors given any thought as to how to extend this work to cover more languages?
>
> Adding a new language requires specific data for that language. ASR data is needed to train the speech tokenizer, and unsupervised speech data is used for the speech decoder. TTS data is required to train the text-to-token model, while text pre-training data is used to synthesize interleaved speech-text data. Finally, dialogue data is necessary to build a spoken chatbot in the target language.
>
> > Are there any plans to open source the tokenizer, text-to-token model, or the speech LM itself?
>
> We plan to release the tokenizer and the speech LM in the camera-ready revision if accpected.
>
> > Also, it would be nice if the authors could describe the amount of computation required to pretrain the speech LM.
>
> We pretrain our 9B LLM on a total of 1 trillion tokens, which corresponds to approximately 5.4e22 FLOPs.

---

### Official Review · Reviewer_9jmQ · 2024-11-03

**Soundness:** 3
**Presentation:** 2
**Contribution:** 3
**Rating:** 6
**Confidence:** 2

**Summary:**

The paper proposes a speech-text pretraining process for scaling speech-language model training without acquiring large amounts of speech audio data. The process mainly includes an ASR-based low-bitrate speech tokenizer and a text-to-speech-token model to produce large quantities of speech tokens for speech-text pertaining. The pre-trained model is fine-tuned on spoken dialog datasets and shows competitive performance compared to existing SOTA models.

**Strengths:**

1. The ASR-based speech tokenizer achieves semantic information preservation and decent speech audio reproduction at the same time.
2. The low-bitrate speech tokenizer and the text-to-token model effectively use the existing large amounts of text data to synthesize large amounts of speech tokens, which saves resources to collect large amounts of speech audio data and improves the language model's speech performance after pretraining.

**Weaknesses:**

The weaknesses are mainly in terms of paper writing and presentation.
1. The paper mentions "we are first to use supervised semantic tokens for SpeechLMs". However, one of the baselines, Mini-Omini also uses a whisper-based speech tokenizer.
2. The details on how the speech and text modalities are interleaved are missing.
3. As an important part of the process, the details of the text-to-token model are missing—for example, model architectures, training schemes, etc.
4. The large amounts of speech tokens generated by the text-to-token model are still from existing datasets and speech-synthesized audio from text. How is this process different from generating speech tokens from synthesized speech audio using large amounts of text? For example, llama-omni also uses cosy-voice to synthesize speech audio to augment training data. What's the innovation here between text-to-speech-to-token and text-to-token?

**Questions:**

See weaknesses.
How are the speech and text tokens interleaved to form training samples? What are the details of this data creation process?
How does the model benefit from interleaved speech and text modalities?
How do you deal with the different sampling rates and information granularities between speech and text tokens during the process?

---

> ### Author Response · Authors · 2024-11-22
> **Response to Reviewer 9jmQ (1/2)**
>
> Thank you for highlighting the weaknesses in our work and providing valuable suggestions to improve our writing and presentation. Based on your feedback, we have made several improvements and update our paper. Specifically, we have added more details about the interleaved data construction process (Cf. Section 2.2) and better established this concept in the abstract and introduction. Additionally, we have included detailed training data statistics (Cf. Appendix A) and provided comprehensive training details for our model (Cf. Appendix B). **We hope these changes address your concerns, and if so, we would be grateful if you would consider raising your score.** Thank you again for your valuable feedback!
>
> > The paper mentions "we are first to use supervised semantic tokens for SpeechLMs". However, one of the baselines, Mini-Omni also uses a whisper-based speech tokenizer.
>
> Mini-Omni [1] employs different speech representations for input and output. Specifically, it utilizes the encoder from Whisper-small to generate continuous speech embeddings for input, while adopting reconstruction-based discrete tokens from SNAC [2] for output. This inconsistency in speech representation prevents Mini-Omni from performing speech pretraining in large unsupervised speech corpus. In contrast, our approach uses a unified supervised discrete speech token for both input and output, which is better suited for speech pretraining.
>
> [1] Xie, Zhifei, and Changqiao Wu. "Mini-omni: Language models can hear, talk while thinking in streaming." arXiv preprint arXiv:2408.16725 (2024).
>
> [2] Siuzdak, Hubert, Florian Grötschla, and Luca A. Lanzendörfer. "SNAC: Multi-Scale Neural Audio Codec." arXiv preprint arXiv:2410.14411 (2024).
>
> > The details on how the speech and text modalities are interleaved are missing.
> > As an important part of the process, the details of the text-to-token model are missing—for example, model architectures, training schemes, etc.
>
> We tokenize both modalities into discrete tokens and speech and text are interleaved at word level. We additionally include samples for interleaved data on Appendix D.3 for better understanding. More details have been provided in Section 2.2 for constructing interleaved data. We also have add the training data statistics (Cf. Appendix A.3) and training details of text-to-token model (Cf. Appendix B.3). Besides, we also include a evaluation of the text-to-token model (Cf. Section 2.2).
>
> > The large amounts of speech tokens generated by the text-to-token model are still from existing datasets and speech-synthesized audio from text. How is this process different from generating speech tokens from synthesized speech audio using large amounts of text? For example, llama-omni also uses cosy-voice to synthesize speech audio to augment training data. What's the innovation here between text-to-speech-to-token and text-to-token?
>
> The innovation can be summarize as follows:
>
> - **Efficiency**:
> Llama-Omni uses CosyVoice to synthesize speech audio for instruction data, generating only 200K samples. However, synthesizing hundreds of billions of tokens using a text-to-speech-to-token pipeline is highly inefficient. In contrast, we train a text-to-token LLM and leverage the SGLang framework for efficient and scalable synthesis. To demonstrate the efficiency, we conducted an experiment comparing the generation speed of the two methods on an H800 GPU. **The results show that our approach is 70x faster**.
>
>     | Method | Speed (tokens/s/GPU) |
>     | ---- | ---- |
>     | CosyVoice (text-to-speech-to-token) | 360 |
>     | Ours (text-to-token) | 25000 |
>
> - **Simplicity**: CosyVoice uses a text-to-token language model to generate speech tokens from text and a speech decoder to reconstruct speech from these tokens. By directly using a text-to-token LLM to generate speech tokens, we simplify the pipeline, improve efficiency, and avoid accumulated errors introduced compared to the text-to-speech-to-token pipeline.

---

> > ### Author Response · Authors · 2024-11-22
> > **Response to Reviewer 9jmQ (2/2)**
> >
> > > How does the model benefit from interleaved speech and text modalities?
> >
> > The ablation study in Table 5 demonstrates the benefits of interleaved data. For instance, the 9B model trained with 0B interleaved data achieves a score of 2.3 on Llama Questions, compared to 50.7 for the model trained with 600B interleaved data. Furthermore, Figure 3(c) illustrates that adding more interleaved data consistently improves performance in spoken chatbot evaluation.
> >
> > > How do you deal with the different sampling rates and information granularities between speech and text tokens during the process?
> >
> > We conducted an ablation study on the span corruption ratio $\eta$ using the 12.5 Hz tokenizer to determine the optimal value based on evaluation results (see Figure 3(b)). The study revealed that $\eta=0.3$ achieves the best performance across benchmarks. For the ablation study on different sampling rates, we used the same $\eta=0.3$ to sample text spans, resulting in varying speech-to-text token ratios depending on the tokenizer employed. Detailed statistics on the token distribution in the interleaved speech-text data are provided in Appendix A.1.

---

> ### Author Response · Authors · 2024-11-27
>
> We greatly appreciate the time and effort you have dedicated to reviewing our work. We hope that all the points raised in your feedback have been adequately addressed. If there are any remaining concerns or areas requiring further clarification, please do not hesitate to let us know—we would be happy to provide additional details or explanations.

---

### Meta-Review · Area_Chair_LoDt · 2024-12-18

**Metareview:**

This paper introduces a new speech-text pretraining method without large-scale audio-text pre-training data by leveraging interleaved text-to-token data from text corpus.

As the importance of multimodal agentic AI increases, speech-text multimodal training is becoming significant and popular to make better agentic AIs. The proposed method using interleaved data address lack of speech-text parallel corpus, which is the main obstacle of speech-LMs. The method design seems effective and novel.

Althoubh some reviewers pointed out lack of details, those are not main concerns.

With positive scores from all reviewers, AC recommends accepting this paper.

Following the reviewers' comments, AC asks the authors to refine the manuscript to be more clear.

**Additional Comments On Reviewer Discussion:**

The initial scores were 5, 8, 8, 5, and 8.

Major conerns of two reviewers with the borderline rejection scores were wrting-presentation qaulity and unclear experimental description.

For the rebuttal period, it seems that the authors successfully convinced 9jmQ even if XoEC did not changed.

Final scores were 6, 8, 8, 5, and 8.

---

### Decision · Program_Chairs · 2025-01-22

Accept (Poster)